# The Mesenchymal Niche in Myelodysplastic Syndromes

**DOI:** 10.3390/diagnostics12071639

**Published:** 2022-07-05

**Authors:** Chloé Friedrich, Olivier Kosmider

**Affiliations:** INSERM U1016, Institut Cochin, Université de Paris Cité, F-75014 Paris, France; olivier.kosmider@aphp.fr

**Keywords:** niche, hematopoietic stem cell, mesenchymal stromal cells, models

## Abstract

Myelodysplastic syndromes (MDSs) are clonal disorders characterized by ineffective hematopoiesis, resulting in cytopenias and a risk of developing acute myeloid leukemia. In addition to mutations affecting hematopoietic stem cells (HSCs), numerous studies have highlighted the role of the bone marrow microenvironment (BMME) in the development of MDSs. The mesenchymal niche represents a key component of the BMME. In this review, we discuss the role of the mesenchymal niche in the pathophysiology of MDS and provide an overview of currently available in vitro and in vivo models that can be used to study the effects of the mesenchymal niche on HSCs.

## 1. Introduction

Hematopoiesis (HP) is the process by which the pool of all mature blood cells is replenished throughout an organism’s life time [1]. The main player of this process is the hematopoietic stem cell (HSC), which has the ability to self-renew and differentiate, giving rise to all blood cell types [2]. Under physiological conditions, HSCs are located in a specialized niche of the bone marrow microenvironment (BMME), called the “HSC niche”, which is essential for their maintenance and functional activity. Disruptions in the HSC niche inevitably affect normal hematopoiesis and are thought to contribute to the emergence of hematopoietic malignancies in both mice and human. Myelodysplastic syndrome (MDS) is a clonal disorder of hematopoiesis characterized by marrow failure and a high propensity to progress to acute myeloid leukemia. This disease is due to complex combinations of changes in the HSC genome that result in heterogeneity in both clinical phenotype and disease outcome. In recent years, accumulating evidences have suggested that BMME may act as a key mediator providing a ‘fertile inflamed’ environment where interactions between components resulted in disease homeostasis. In other words, specific alterations in the HSC niche could act as predisposing events, facilitating the survival and expansion of mutant hematopoietic cells.

In this review, we will discuss the role of the HSC niche, especially the mesenchymal niche, in the pathophysiology of MDS and provide an overview of currently available in vitro and in vivo models to study the effects of the mesenchymal niche on MDS HSCs.

## 2. The HSC Niche

The idea that HSCs occupy a specialized niche in hematopoietic tissues was first refined by Schofield in 1978 [3]. An anatomical description of the HSC niche has been refined in recent years with the introduction of SLAM family receptors [4] and technical breakthroughs associated with a series of elegant genetic ablations in mouse models. The HSC niche is now considered a complex multicellular network, composed of both HSCs and their progeny (HSPCs), which are functionally and molecularly heterogeneous [5], and non-hematopoietic cells, that contribute to the localization, maintenance and differentiation of HSC in redundant manner [6,7,8]. For simplicity, we decided to divide the HSC niche into two anatomically compartments: the endosteal niche and the central niche (Figure 1). The endosteal niche is located in close proximity to the bone surface. This region is lined with cells of the osteoblastic lineage, of mesenchymal origin, in various stages of differentiation. Osteoclasts, of hematopoietic origin, involved in bone resorption, are also present in this region and participate in bone metabolism in balance with osteoblasts. Only a small percentage of HSCs (15%) are localized at the endosteum level, but several studies have shown that the endosteal niche plays an important role in the regulation of hematopoiesis. In the 1990s, Taichman et al. established in vitro co-culture systems showing that osteoblasts could support primitive hematopoietic cells via the production of granulocyte colony-stimulating factor (G-CSF) [9]. Since then, several murine models have confirmed the link between hematopoiesis and osteoblasts in vivo [7,10,11,12]. Several osteoblastic signals have thus been described to promote HSC quiescence, which is essential to preserve the self –renewal capacity of HSCs [13,14]. The central niche contains the majority of sinusoids and arterioles. Under homeostatic conditions, approximately 85% of HSCs are in close contact with sinusoidal blood vessels. Bone marrow endothelial cells (BMECs) play a crucial role in the maintenance of HSCs: they form a network of blood vessels that provide angiocrine signals regulating HSC development and homeostasis [15,16]. Under pathological conditions, BMECs also play an important role in the development and progression of diseases [17]. Indeed, BMECs have been shown to be a fundamental source of angiocrine factors that promote the development and survival of T-cell acute lymphoblastic leukemia (ALL) in the BM niche [18]. In addition, BMECs have been shown to be strongly disrupted by the presence of acute myeloid leukemia (AML) [19,20].

In both endosteum and central BM, HSCs reside in perivascular niches, where they are critically maintained and regulated by BMECs and their associated perivascular mesenchymal stem cells (MSCs) [6]. For simplicity, all of these perivascular MSCs are grouped under the term ‘mesenchymal niche’.

## 3. The Mesenchymal Niche

MSCs were first described in the 1960s and 1970s by Friedenstein’s group as a population present in the rodent BM that has the ability to form fibroblastoid colonies (CFU-F) when cultured on plastic, to make bone, and to reconstitute the hematopoietic microenvironment when transplanted subcutaneously [21]. Because the definition of MSCs has long been debated, in 2006 the International Society for Cellular Therapy (ISCT) formulated minimal criteria for defining MSCs to create a broader consensus for a more uniform characterization of these cells. These include adherence to plastic under standard culture conditions; expression of CD105, CD73 and CD90; the lack of expression of CD45, CD34, CD14 or CD11b, and CD79α or CD19; and HLA-DR surface molecules as well as the ability to differentiate into osteogenic, chondrogenic, and adipogenic lineages in vitro [22]. Over the past 20 years, technological advances in genetically modified animals, in addition to cell labeling, have led to a better understanding of the phenotypic and functional heterogeneity of MSCs. In vivo, MSCs are crucial components of BMME. Although rare (0.001–0.01% of total BM mononuclear cells), they tightly control the fate of HSCs by direct interaction or paracrine secretion of soluble factors. Murine Nestin-GFP+ MSCs were first described as perivascular MSCs in close contact with HSCs and HSPCs. In an enzymatically digested murine BM, Nestin-GFP+ cells were clonogenic and demonstrated the ability to form mesenspheres as well as a robust self-renewal capacity, reflected by serial transplantation in vivo [6]. In addition, they expressed several HSPC maintenance factors, such as CXCL12, stem cell factor (SCF), angiopoietin-1, IL-7 and vascular cell adhesion molecule 1 (VCAM1) [23]. Interestingly, Nestin-GFP+ MSCs showed several similarities to the more recently identified CXCL12-abundant reticular cells (CAR) associated with sinusoidal endothelium [14]. CAR cells are the main producers of CXCL12 and SCF [14]. The reduction in the HSPC pool upon short-term ablation of CAR cells underscored the fundamental role of these cells in maintaining primitive HSPCs in a mouse model. Using SCF-GFP knock-in mice, Ding et al. found that expression of the adipo-osteogenic regulator of leptin (Lepr) was strongly restricted to perivascular stromal cells expressing SCF-GFP [15]. SCF deletion of Lepr-expressing perivascular stromal cells also resulted in HSC depletion in BM.

All of these results support the concept of a mesenchymal niche composed of different subpopulations of MSCs, located in interconnected areas. Using single-cell RNA sequencing, Wolock et al. confirmed this heterogeneity by reconstructing the transcriptional hierarchy of mouse bone MSC states, thereby providing key information about transcriptional events that direct osteoblast, chondrocyte, and adipocyte differentiation from stromal stem cells [24].

In addition to controlling HSC fate, several studies have demonstrated that MSCs have broad and potent immunoregulatory properties, both in vitro and in vivo. MSCs have been shown to influence adaptive cells, such as T and B cells, and innate immune cells, such as dentritic cells, natural killer (NK) cells, monocytes, and macrophages by secreting several factors such as indoleamine 2,3-dioxygenase (IDO) and various cytokines [25]. These soluble factors released by MSCs are grouped under the term of ‘Secretome’.

As a final feature, Sacchetti et al. reported for the first time that CD146-label human BM cells residing in the sinusoidal wall, which were enriched for CFU-F activity, were able to maintain hematopoietic activity when transplanted as heterotopic ossicles into immunodeficient mice, thereby transferring a hematopoietic microenvironment [26]. This particular property of MSCs to form an organized microenvironment has led some teams generating bone organoids (defined hereafter as ossicles), that mimic the MSC niche, in which stromal cells and active hematopoiesis can be detected [27,28].

## 4. The Mesenchymal Niche in MDS

As previously mentioned, the mesenchymal niche supports hematopoietic stem cells and their progeny. It is therefore reasonable to hypothesize that a disruption of the mesenchymal niche may contribute to the pathogenesis of MDS. MDS is a clonal hematopoietic disorder, characterized by ineffective hematopoiesis with varying degrees of dysplasia and peripheral cytopenias and a risk of progression to acute myeloid leukemia (AML). There exist many subtypes of MDSs according to the 2016 WHO classification (Appendix A) and a relatively heterogeneous spectrum of presentation.

The pathogenesis of MDS involves an initial ancestral lesion, followed by the progressive acquisition of subsequent somatic mutations, resulting in a highly diverse clonal hierarchy [29]. While driver mutations can lead to de novo disease formation, some MDSs result from the expansion of an initially minor clone, called clonal hematopoiesis (CH), present in the blood of some otherwise healthy individuals [30]. We can therefore ask how minor mutated clones could persist in the BM and eventually lead to clonal outgrowth. Emerging evidence indicates that the mesenchymal niche plays a critical role in the initiation and progression of MDS. MDS is thought to be not only a disease of hematopoiesis but also of the surrounding mesenchymal niche. On the basis of previous studies, two non-mutually exclusive hypotheses have therefore been proposed to explain the role of the mesenchymal niche in the pathophysiology of MDSs: (i) the niche-induced leukemogenesis model, in which niche cell abnormalities act as an initiating event leading to or predisposing to the development of a hematopoietic malignancy; (ii) the niche-facilitated leukemogenesis model, in which leukemic or pre-leukemic cells remodel the niche to create an environment facilitating their expansion and survival. These two models are summarized in Figure 2.

### 4.1. The Niche-Induced Leukemogenesis Model

This hypothesis was first supported by the description of donor cell leukemia, defined as oncogenic transformation of apparently normal transplanted donor cells in the allogenic patient’s environment, but not in the donor’s environment [31], presumably due to an altered microenvironment in the transplant recipients. Thus, several studies in mice have suggested that MDS may originate from the niche itself. The first experimental evidence that a specific and distinct stromal cell type can initiate BM failure came from a study in which the disruption of *Dicer1*, an endonuclease essential for microRNA biogenesis, in immature osteoprogenitors but not in mature osteoblasts, led to MDS-like disease. *Dicer1*-deletion in immature osteoprogenitors also down-regulated *Sbds*, the gene responsible for Schwachman-Diamond syndrome (SDS), a human disease characterized by bone marrow failure and predisposition to leukemia [32]. Despite these interesting observations in mice, the role of *Dicer1* in the MDS pathophysiology remains debated: in a recently published study, Vasta et al. showed that no patient with a pathogenic germline variant of *Dicer1* developed MDS or leukemia [33]. On the contrary, consistent with what was previously observed in mice, MSCs from MDS patients were shown to abnormally express low levels of *Dicer1* and *Sbds* mRNA compared to controls [34]. Moreover, Zambetti et al. demonstrated that MSCs lacking *Sbds* also activate p53 and release the molecules S100A8 and S100A9, which cause genotoxic stress in HSPCs, potentially promoting leukemic transformation [35].

The concept of niche-induced oncogenesis is also supported by other studies indicating the disruption of the BM architecture in MDS, particularly the role of MSCs themselves as a major component of this disrupt architecture [36,37]. Indeed, it was shown that MSCs from MDS patients exhibited cytogenetic, transcriptomic and epigenetic abnormalities, increased replicative stress and shorter telomeres compared to healthy controls [36,38,39,40,41]. MSCs from MDS patients also exhibited increased senescence, impaired proliferation capacity and increased inflammatory phenotypes [35,36,42,43]. However, the clonal origin of MDS-MSCs still remains controversial. Indeed, in their study, Jann et al. suggested that mutations and/or cytogenetic abnormalities in MSCs may be associated with expansion in culture but not present in primary BM [38].

In addition to their intrinsic abnormalities, numerous studies have demonstrated an effect of MDS-MSCs on HSPCs and on their immune environment via the secretion of soluble factors. Indeed, MDS-MSCs display reduced expression of essential HSC niche factors like CXCL12, ANGPT1 and KITL and a reduced ability to support HSPCs [41]. Furthermore, activating mutation of the *Ptpn11* gene in mouse BM MSCs has been shown to promote the development and progression of the MDS/MPN juvenile myelomonocytic leukemia (JMML) through excessive production of the CC chemokine CCL3, which recruits monocytes to the area where HSCs also reside and activates them [44]. MDS-MSCs have also been shown to be able to reprogram healthy HSPCs. In co-culture models, healthy HSPCs co-cultured with MDS-MSCs failed to engraft into primary and secondary xenograft recipients [41].

All of these observations seem to confirm an active role of MSCs in the pathogenesis of MDS. However, we can question whether MDS-MSCs are primary dysfunctional or whether it is a secondary effect induced by the MDS HSPCs. This consideration now leads us to discuss the niche-facilitated leukemogenesis model, in which MDS HSPCs can remodel the ME to their advantage.

### 4.2. The Niche-Facilitated Leukemogenesis Model: The Fundamental Role of Inflammation

Somatic mutations are known to occur in HSPCs during normal DNA replication. With aging, “inflamm-aging” is widely reported. This includes elevated circulating levels of pro-inflammatory cytokines such as TNFα, IL-1β, IL-6 [45]. Due to this “inflamm-aging” and genomic instability, HSPCs acquire certain mutations in epigenetic regulators such as *DNMT3A*, *TET2*, and *ASXL1* genes, defining the clonal hematopoiesis (CH) that represents the first step in MDS pathogenesis [46]. However, most of these molecular alterations are not sufficient to explain the increased self-renewal potential of HSPCs in MDSresponsible for clonal dominance and disease progression. Thus, an inflamed microenvironment has been posited to favor the expansion of HSCs with CH mutations, providing an explanation for the clonal dominance of mutant HCSs over the normal HSC population in the context of aging. In their study, Muto et al. found that chronic inflammation was a determinant for the competitive advantage of MDS HSPCs and for disease progression. They found that MDS HSPCs had an altered response to chronic inflammation and that this response, based on the non-canonical NF-kB pathway, contributed to the sustained myeloid expansion of these cells and their competitive advantage compared to normal HSPC [47]. Using a mouse chimera model, Hormaechea et al. tested the hypothesis that infection could drive *DNMT3A* CH, and they found that IFNγ signaling, induced during chronic infection, resulted in loss of *DNMT3A* CH function, providing definitive evidence that an altered microenvironment can act on clonal hematopoiesis [48].

An inflamed ME is now thought to contribute to clonal expansion of HSCs carrying somatic mutations. There is also evidence showing that MDS HSPCs can themselves remodel the stem cell niche by creating an inflammatory environment that they are resistant to. For example, in a number of murine models of CH or MDS, such as the TET2^−/−^ and 5q^−^ MDS models, we observe an increase in alarmins (S100A8, S100A9), with the activation of the inflammasome NLRP3, which directs the generation of IL1β, S100A9 and ROS, creating a proinflammatory environment [49,50]. Meydyouf et al. demonstrated in a co-culture model that HSPCs from MDS were able to reprogram healthy MSCs. Furthermore, in patient-derived xenograft experiments, they showed that primary MDS samples were dependent on interaction with BM-derived MSCs for propagation. This led to the hypothesis that HSPCs from MDS could re-educate their BMME to create a favorable environment for their preferential growth [51].

Taken together, these data indicate that the local inflammatory niche may act as an “amplification loop”, which (1) could lead to the selection of pre-existing clonal hematopoiesis and (2) may accelerate the acquisition of additional genetic events facilitating the transformation to MDS.

Whether primiraly or secondarily induced, these studies indicate that an inflamed “mutagenic” ME driven by mesenchymal niche leads to disruption of HSPC homeostasis, facilitating somatic mutations, selectionand subsequent MDS clonal expansion. In this context, the mesenchymal counterpart appears essential to studying the pathophysiology of MDS, which appears to be not only a hematopoietic clonal disorder but also a tissue disorder. In the last section of this review, we discuss the current models used to study the impact of the mesenchymal niche on the pathophysiology of MDS.

## 5. In Vitro and In Vivo MSCs Based Models

### 5.1. In Vitro Models

To study the role of the mesenchymal niche on HSPCs, in vitro 2D co-culture models have been historically used. In a simplified way, these models are based on the use of a feeder layer of MSCs that is in contact with the HSPCs. These models are useful for studying the characteristics of MDS-MSCs in terms of morphology, proliferation rate and ability to maintain the potential of HSPCs under co-culture condition [36,41,42,52]. Moreover, these models allow us to study the secretome of MSCs in a simple way. Although easy to use, well established, fast and inexpensive, in vitro 2D co-culture models have some limitations: in the long term, HSCs become exhausted and lose their stemness potential. In addition, these models lack some fundamental features of the BMME such as niche-derived paracrine factors, extracellular matrix (ECM) components, and they do not take into account the physical properties of BMME, such as stiffness, oxygen, other contacting cells and the vascular system.

In recent years, new in vitro models have been developed to overcome the shortcomings of traditional 2D cell culture methods. Three-dimensional in vitro models have been shown to recreate the natural ME of tumor cells more accurately than 2D models. Allowing cells to grow in a 3D environment, e.g., spheroids, reformed culturing processes of HSPCs. Although there is growing evidence that in vitro 3D models can be useful for improving preclinical research, many 3D models have been shown to be less effective, easier and more reproducible than 2D models [53,54]. Bioengineered in vitro models, such as organs-on-a-chip supported by cells of mesenchymal origin, are bringing researchers closer and closer to mimicking complex in vivo environments [55]. Although promising, organs-on-a-chip do not yet fully recapitulate the complexity of bone marrow and do not allow the study of HSC homing [56]. To overcome these limitations, in vivo models are still needed.

### 5.2. In Vivo Models

#### 5.2.1. Xenotransplantation Models

Xenotransplantation models (PDX models) are used to establish hematological malignancies in immunodeficient mice. In recent years, non-obese diabetic NOD/SCID and NOD/SCID IL2rɣ^−/−^ (NSG) strains have provided insight into normal human stem cell biology, the cancer stem cell concept [57], leukemic clonal heterogeneity [58] and clonal hierarchy [59]. However, xenotransplantation of primary human MDS cells into immunodeficient mice has had limited success with a poor efficacity and a transient engraftment skewing toward the lymphoid lineage [60,61,62]. Furthermore, this engraftment was often supported by residual normal cells and not by the MDS clones [63].

To make a long story short, cumulative work has produced a total of only approximately 100 MDS PDXs over the past 30 years [64]. This low engraftment could be related to the dependence of MDS cells on human-specific factors that are not provided by the mouse ME. In order to improve the MDS-HSPC engraftment and to provide specific human factors necessary for HSCs maintenance, successive modifications of the mouse genome by introducing human genes coding for critical human signaling molecules have been performed. The recent development of humanized immunodeficient ‘MISTRG’ mice expressing human **M**-CSF, **I**L-3/GM-CSF, **S**IRPα, **T**hrombopoietin in the **R**ag^−/−^, IL2r**ɣ**^−/−^genetic background from their endogenous murine loci has allowed MDS cells to engraft efficiently and give rise to multilineage hematopoiesis [65]. The main limitation of this model is the reported development of anemia, which is also a feature of human MDS. Furthermore, intrahepatic injection in newborn pups not only poses logistical problems but could also potentially influence tumor behavior, as this system is a ‘young’ niche, in contrast to the BM niche of elderly MDS patients. Another major drawback remains with this model: incomplete cross-compatibility between the murine stroma and transplanted human hematopoietic cells. Notably, mouse bones do not reflect the human BMME, precluding the possibility of studying species-specific functional interactions between human cancer cells and their associated niches. It would be essential to be able to study these interactions under more humanized conditions to better understand the crosstalk between HSPCs and MSCs, which we predict may be critical in the initiation, maintenance and progression of MDS.

We could imagine manipulating the mouse ME to mimic the human BM niche and thus provide functional support to human stem cell. The first and simplest approach consist ofco-injecting patient-derived autologous/allogeneic MSCs with MDS cells (CD34+ BM cells or CD3-depleted BM cells) directly into the medullary cavity of sublethally irradiated NSG mice [51,66]. This strategy has yielded variable results in terms of promoting engraftment of MDS samples: Medyouf et al. showed that co-injection of MDS CD34+ HSPC with their corresponding in vitro–expanded MSCs into the medullary cavity of NSG mice significantly increased reconstitution with human MDS cells compared to transplantation of CD34 alone or co-injection with MSCs from age-matched normal BM [51]. Rouault Pierre et al. did not obtain the same results: no difference was observed between mice that received CD34+ cells alone or with MSCs. Moreover, intrabone-injected MDS MSCs did not persist for >1 week in NSG mice [66]. Thus, these results suggest that presence of hMSCs in the murine bone marrow cavity is limited over time and fail to establish a long-term human-like ME (Figure 3A). Other transplantation methods such as the use of a three-dimensional scaffold may be useful forfurther dissecting the role of MSCs in supporting the MDS clone.

#### 5.2.2. Humanized Bone Marrow-like Structures

Over the past decade, several groups have developed methods to model the human BM microenvironment in mice using subcutaneous humanized marrow-like structures (hBMLS) [27,28,67,68]. In summary, all of these approaches are based on the use of scaffolds of different origins: osteoinductive ceramic [69], gelfoam [28], extracellular matrix (ECM)-derived gels [27] or human bone substance combined with gelfoam [67]. These scaffolds are seeded with primary MSCs, which have been previously expanded in vitro. As previously discussed, MSCs have the ability to differentiate into bone, adipocytes and various others stromal components [22]. Thus, these properties allow MSCs to create a stromal layer on the established scaffold, which provides a “niche like units” in which other cells can reside. In some protocols, seeded MSCs are first differentiated in vitro into cartilage before being implanted in a process called chrondrogenic priming [70]. Growth factors such as BMP-2 or BMP-7 (bone morphogenetic proteins) can also be added to stimulate bone formation [71]. In most protocols, human hematopoietic cells (BM CD34+ cells or BM CD3 depleted CMN) are then injected directly into the scaffolds, which are then introduced subcutaneously into non-irradiated immunocompromised mice. Depending on the protocols, 2 to 6 scaffolds can be introduced per animal. After a few weeks of development, structures called “ossicles” are formed mimicking a human BMME, comprising bone, cartilage, and MSCs of human origin. These humanized ossicles are therefore colonized by the mouse endothelium. In Reinisch’s protocol, human hematopoietic cells are injected 8-10 weeks post BM-MSC application in mice, directly onto the scaffold or by intravenous injection, after busulfan treatment or sublethal irradiation. Daily PTH treatment can also be added in order to stimulate osteoblast formation and increase BM MSCs engraftment [2,27]. Long term hematopoiesis is observed after 12–32 weeks depending on the protocol (Figure 3B).

These models have been shown to allow higher and more stable engraftment rates of healthy or leukemic HSPC, including favorable risk AML samples that are notoriously difficult to engraft in NSG mouse strains. In AML, the presence of humanized microenvironment promotes engraftment of AML patients’ sample and HSC self-renewal. Moreover, clonal heterogeneity as observed in patients is much better preserved in the human ME than in mouse BM [69]. Humanized ossicles facilitate robust engraftment of PML-RARA positive human acute promyelocytic leukemia (APL) cells and primary human myelofibrosis (MF) cells, both of which are difficult to engraft in conventional xenotransplants. Furthermore, with humanized niche, the frequency of leukemia-initiating cells (LIC) is significantly higher than that determined in NSG mouse BM with the same sample, suggesting that AML LICs frequency is critically dependent on interactions between leukemic cells and the niche [27].

With regard to MDS, only a few studies have been published to date. In a recent work, Mian et al. have shown with a gelfoam approach, a level of engraftment previously unattainable in mice [68]. Created from MDS-MSCs seeded on scaffolds, extramedullary niches allowed the engraftment of HPSCs from approximately 94% of MDS, regardless of MDS subtype. They observed persistent long-term engraftments of human CD45+ cells in the scaffold ranging from 0.2 to 86%, with 82% of cases having >20% human CD45+ cells. In addition, they were able to functionally demonstrate HSC self-renewal and differentiation capacity within the niche using secondary transplant assays. Elegantly, they showed that MDS-HSPC were able to migrate out of the primary niche but were subsequently homing and engrafting only in niches seeded with human MSCs, revealing the dependence of MDS-HSPC on their ME. Similar results were found by Altrock et al.: in their study, they interrogated the feasibility of different ossicle models for application with primary MDS samples (gelfoam, ECM-derived gels or a combination of human bone and gelfoam). They showed that HSPCs and MSCs derived from MDS, which did not engraft significantly in NSG mice after intrafemoral (IF) co-injection, were capable of colonizing humanized scaffold models [67]. In this study, ossicle-based protocols using gelfoam or a combination of human bone and gelfoam have resulted in significantly higher graft rates. Overall, these studies suggest that humanized bone marrow-like structures appear to be the most relevant models for studying MDS pathogenesis and leukemia cell- niche interactions. These models also highlight the possibility to genetically modify MSCs to study the role of specific signaling pathways in these humanized hematopoietic microenvironments.

However, compared to other hematological malignancies, the hematopoietic stem cell engraftment rate in these models remains modest, leading us to consider further areas of improvement. The limitations of the ossicle models lie principally in their chimeric nature, including vasculature of mouse origin. The importance of the vascular system in BM has been widely demonstrated by in vivo animal studies: it provides distinct signaling supporting HSC, as well as stromal cell type [13,72]. Different research teams have already generated such models, harboring human vasculature, construct from primary endothelial cells (ECs) or immortalized cells like HUVEC cell lines [73,74,75]. In these models, EC and MSCs are simultaneously implanted in a gel or other carrier material and interact closely during neo-vascularization, thereby improving the vascular tissue formation. It could be then interesting to apply this strategy to ossicles.

## 6. Conclusions

Treatment of high risk MDS is dominated by monotherapy with hypomethylating agents (azacytidine and decitabine), but new combinations are being evaluated in clinical trials [76,77,78]. Stem cell transplantation remains the only curative treatment but is associated with a high risk of transplant-related mortality and relapse. Agents that stimulate erythropoiesis continue to be first-line treatment for the low-risk MDS, and Luspatercept, which acts as TGF ligand trap, has shown promise as second-line treatment for sideroblastic MDS [79]. Despite these advances, some patients remain resistant to treatment or eventually relapse. As with AML, this may be due to the presence of MDS-initiating cells (MDS-IC) that persist after treatment. Indeed, there is increasing evidence linking stemness to prognosis and therapy failure, particularly the dependence of MDS-ICs on niche cells. It is therefore of utmost importance to dissect the interactions existing between MDS-ICs and their ME. For this purpose, humanized bone marrow-like structures are the ideal playground: MSCs from different origins can be used to construct a “custom” humanized ME, that is capable of hosting HSC-MDS cells or normal hematopoiesis. In this context, the use of single cell RNA data would be an interesting approach to explore transcriptional differences existing between normal stem cells and MDS-ICs in normal or pathological ossicles.

## Figures and Tables

**Figure 1 diagnostics-12-01639-f001:**
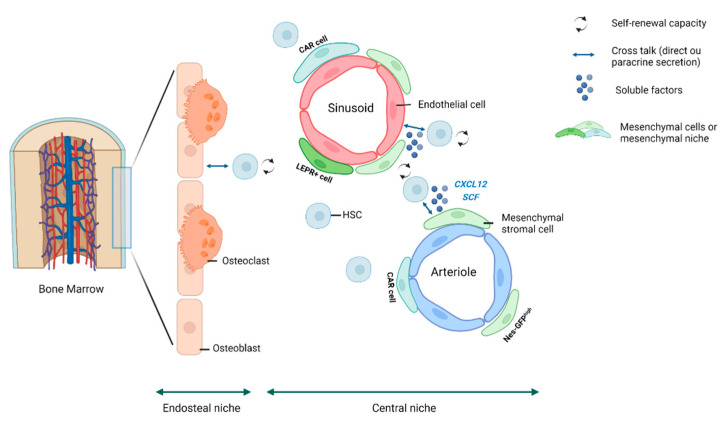
The bone marrow HSC niche. Schematic representation of the hematopoietic stem cell niche, divided into two main compartments: the endosteal niche and the central niche. Both compartments are composed of HSCs surrounded by other cells that regulate their fate. Mesenchymal cells, shown in green, represent one of the main components that regulate HSCs. Because of the heterogeneity of MSCs, we group them under the term “mesenchymal niche”.

**Figure 2 diagnostics-12-01639-f002:**
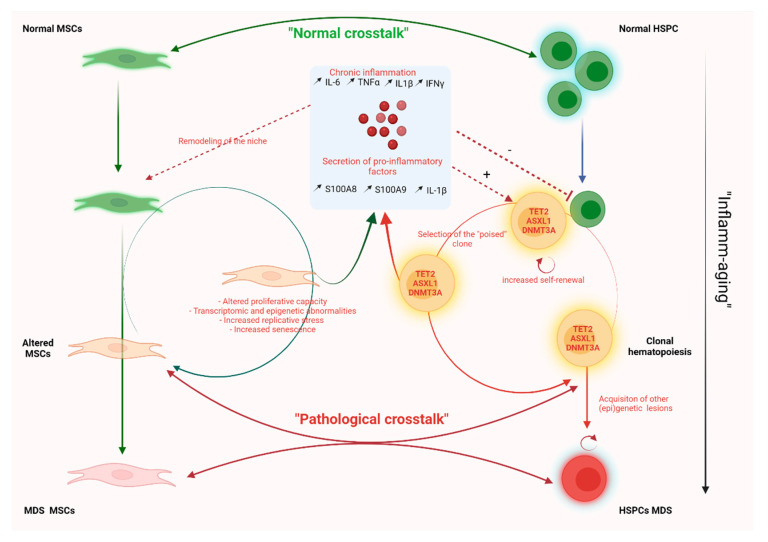
Proposed model explaining the interaction of MSCs and HSPCs in the pathophysiology of MDS. Physiologically, there is a dialogue between normal HSPCs and normal MSCs. During aging, the BMME becomes more inflammatory (“inflamm-aging”) and will favor the expansion of minority clones carrying CHIP mutations (*TET2*, *ASLX1*, *DNMT3A*). In turn, these selected clones will secrete proinflammatory molecules that will remodel the niche. The “reprogramed” MSCs will in turn secrete pro-inflammatory factors. This pathological crosstalk will progressively amplify and thus promote the appearance of new mutations within the HSPCs and lead to MDS.

**Figure 3 diagnostics-12-01639-f003:**
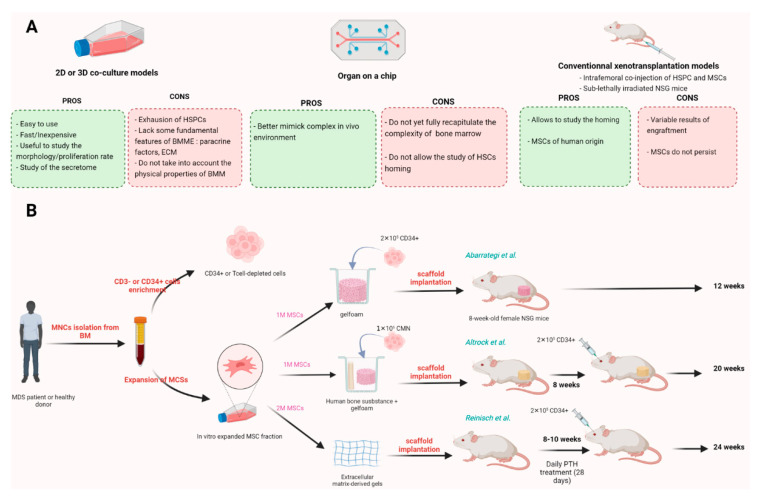
Overview of different approaches to study the effect of the mesenchymal niche on hematopoiesis. (**A**) Historically, 2D and 3D culture models have been used. However, organs on CHIP better model the complexity of the bone marrow. Xenotransplantation models are generally used to study hematological diseases, but the percentage of engraftment is very low when applied to MDS. (**B**) In an effort to improve MDS engraftment, models of humanized bone marrow-like structures have been generated. All these models are based on a preliminary amplification step of primary MSCs, which are then seeded on a scaffold which can be of different origin. The HSPCs are then either implanted directly into the scaffold or injected intravenously into the mouse [27,28,67].

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
