# Peer review of "The Mesenchymal Niche in Myelodysplastic Syndromes"

_diagnostics, 2022, doi:10.3390/diagnostics12071639_

Round 1

Reviewer 1 Report

The authors were right to the point. It is well written and clear for the audience. I recommend its publication.

Author Response

Dear Reviewer 1, 

Thank you for your positive comments. 

Reviewer 2 Report

The manuscript entitled “The Mesenchymal niche in Myelodysplastic syndromes” is an interesting work. However, the following comments should be addressed:

- A figure outlining the hematopoietic niche is missing. (Section 2. The HSC niche)

- Please revise “in vitro” and “in vivo”. These terms should be in italic, but not all are.

- On line 128, the sentence “MDS is also a clonal hematopoietic disorder [30].” is a repeated idea and should be removed.

- A brief description of the figure should be added to Figure 1.

- Clonal hematopoiesis (CH) is written out in full in line 132 (its first appearance). In line 196, it should only be CH. On the other hand, ME (line 210) is not written out in full.

- In section “5.1. In vitro models,” the 3D models to study the hematopoietic niche should be improved.

Reviewer 3 Report

The manuscript is good contribution to the field of Myelodysplastic syndrome.

Please improve the quality of the figures.

Please include one  tables related to Myelodysplastic syndrome and one table related to clinical translation.

Author Response

Please improve the quality of the figures.

We have improved the quality of the figures

Please include one  tables related to Myelodysplastic syndrome and one table related to clinical translation.

We have added in the supplementary data the WHO 2016 classification and made some modifications to the text. As this review is not clinical but more fundamental, we have voluntarily chosen not to develop the clinical part, which would certainly be developed by other reviews.

Round 2

Reviewer 3 Report

No more comments